# Cu(II) Ion Adsorption by Aniline Grafted Chitosan and Its Responsive Fluorescence Properties

**DOI:** 10.3390/molecules25051052

**Published:** 2020-02-26

**Authors:** Bahareh Vafakish, Lee D. Wilson

**Affiliations:** Department of Chemistry, University of Saskatchewan, Saskatoon, SK S7N 5C9, Canada; bav128@mail.usask.ca

**Keywords:** chitosan, grafting, Cu(II) adsorption, adsorption mechanism, fluorescence, in situ sensor

## Abstract

The detection and removal of heavy metal species in aquatic environments is of continued interest to address ongoing efforts in water security. This study was focused on the preparation and characterization of aniline grafted chitosan (CS-Ac-An), and evaluation of its adsorption properties with Cu(II) under variable conditions. Materials characterization provides support for the grafting of aniline onto chitosan, where the kinetic and thermodynamic adsorption properties reveal a notably greater uptake (>20-fold) of Cu(II) relative to chitosan, where the adsorption capacity (*Q_m_*) of CS-Ac-An was 106.6 mg/g. Adsorbent regeneration was demonstrated over multiple adsorption-desorption cycles with good uptake efficiency. CS-Ac-An has a strong fluorescence emission that undergoes prominent quenching at part per billion levels in aqueous solution. The quenching process displays a linear response over variable Cu(II) concentration (0.05–5 mM) that affords reliable detection of low level Cu(II) levels by an in situ “*turn-off*” process. The *tweezer-like* chelation properties of CS-Ac-An with Cu(II) was characterized by complementary spectroscopic methods: IR, NMR, X-ray photoelectron (XPS), and scanning electron microscopy (SEM). The role of synergistic effects are inferred among two types of active adsorption sites: electron rich arene rings and amine groups of chitosan with Cu(II) species to afford a *tweezer-like* binding modality.

## 1. Introduction

Environmental contamination by toxic heavy metal ions is an important issue that relates to uncontrolled contaminant release from industrial activities [1,2,3]. Metal ions are not biodegradable and may cause disease through bioaccumulation in the tissues of living organisms [4]. Copper is among the various metals used in alloys, fertilizers, electrical equipment, plumbing materials, and heat exchangers, whereas it is commonly found as Cu(II) in industrial effluent [5]. Cu(II) is an essential micronutrient for plants, animals, and humans. While Cu(II) is essential for enzymatic activity, there exists a narrow range between copper deficiency and toxicity [6]. The annual world production of copper is increasing and this will likely result in elevated levels of soluble forms of Cu(II) in aquatic environments [7]. Elevated levels of Cu(II) are known to upset children and adults via adverse effects on the kidney and liver [6]. Therefore, a maximum acceptable concentration of copper in drinking water was set at 1.3 mg L^−1^ by the United States Environmental Protection Agency (US-EPA) [8].

The adverse health effects of copper exposure have stimulated research to develop new tools for detecting Cu(II), especially in aquatic environments. Recently, a number of methods have been tested, including photoconductive decay (PCD) [9], spectrophotometric methods [10], inductively coupled plasma (ICP) [11] with optical and mass spectrometry detection, and electrochemical methods [12] to detect low levels of copper. 

Apart from detection, a need exists for the removal of target pollutants of interest, for which many techniques have been used for contaminant removal from wastewater [13]. Among the various physical, chemical, and biological-based remediation approaches, adsorption methods have received extensive attention because of their relative efficiency, facile infrastructure, and low operational cost [14]. The type of the applied adsorbent is a key factor that governs the effectiveness of the adsorption process [15]. Various adsorbent materials have been examined for Cu(II) removal, including modified clays [16], carbon nanotube composites [17], graphene oxide [18], activated carbon [19], and various types of modified chitosan [19,20,21,22]. Chitosan is a biopolymer material derived from deacetylation of chitin with promising potential as a bio-sorbent due to its relatively low cost, its biodegradability, and its synthetic versatility [23]. To improve the removal efficiency and other physicochemical and mechanical properties of chitosan, synthetic modification via grafting and cross-linking has been employed. These synthetic methods result in alteration of the type and number of functional groups at the surface of chitosan, where such modification strategies often yield a greater number of active adsorption sites [24,25].

To gain insight on the mode of adsorbate uptake with an adsorbent, a study of the adsorption mechanism will aide in the design of materials with improved adsorption sites that contribute to greater adsorption capacity and uptake efficiency. Different adsorption mechanisms have been reported, which include electrostatic interactions, chelation, weak molecular interactions, etc. [26], based on the type of functional groups of the adsorbent [27]. Amine and hydroxyl groups of pristine chitosan play a key role in metal adsorption [28], whereas a more multifaceted sorption mechanism applies to modified chitosan [29].

In the present study, a “*tweezer-like*” chitosan-based adsorbent was studied, where the grafting of aniline onto the amine groups of chitosan was reported in an earlier work [30]. Herein, the “*tweezer-like*” adsorbent was used as an effective *turn-off* sensor for the detection of low levels of Cu(II) in aqueous solution. The uptake properties of this modified adsorbent were investigated in the presence of Cu(II) at equilibrium and dynamic conditions, along with adsorbent regeneration over multiple cycles. The mechanism of adsorption was studied by spectroscopy (IR, ^1^H-NMR, XPS) before and after the adsorption process, along with changes in morphology visualized by SEM. This study reports a *first example* of the use of Cu(II) as a chemical agent that triggers responsive “*tweezer-like*” binding behavior for aniline grafted chitosan with a fluorescence sensor and unique adsorption properties in aqueous media.

## 2. Materials and Methods

### 2.1. Materials

Low molecular weight chitosan (≈75%−85% deacetylation), acetaldehyde, aniline, acetic acid, ethanol 95%, methanol, copper sulfate pentahydrate, nitric acid, sodium hydroxide, deuterium oxide (D_2_O), hydrochloric acid, and EDTA (ethylenediaminetetraacetic acid, disodium salt) were purchased from Sigma-Aldrich and used without further purification. Millipore water was used for the preparation of all aqueous samples.

### 2.2. Preparation of a Tweezer-Like Adsorbent (CS-Ac-An)

The synthesis of a “*tweezer-like*” adsorbent was adapted from a previous report with slight modifications [30]. In brief, 0.6 g of chitosan was dissolved in 40 mL of 2% aqueous solution of acetic acid with mixing for 30 min at 23 °C. To the light-yellow solution, acetaldehyde (145 μL, 1.1 mol equivalents relative to the glucosamine monomer subunits) was added with stirring for 30 min. The mixture was raised to pH 5 using 1M NaOH solution and mixed at 23 °C for 4 h before the addition of aniline (260 μL, 1.1 mol equivalent to glucosamine monomer subunits). The reaction mixture was stirred at 70 °C overnight, when a milky solution was formed. The pH of the resulting mixture was set at 7.0–7.2 by gradual addition of NaOH (1 M) solution during vigorous shaking to yield a pink precipitate. They were separated by centrifugation for 10 min at 1000 rpm and washed several times with Millipore water and ethanol 70% solution. Soxhlet extraction was carried out overnight with ACS grade methanol, followed by vacuum oven drying at 50 °C for 8 h, where a product with yellow flakes was obtained. Appendix A represents the proposed structure of CS-Ac-An. 

### 2.3. Instrumentation

The concentration of Cu(II) in solution was analyzed with a flame atomic adsorption (FAA) spectrometer (Perkin-Elmer AAnalyst 400 AA, Germany) with acetylene/air flow of 2.5 mL min^−1^/10 mL min^−1^, respectively. Quantitative analysis at equilibrium conditions used a wavelength of 249 nm, whereas 216 nm was used for kinetic adsorption studies. Stock solutions were diluted before analysis, where a dilution factor was taken into account for determination of the Cu(II) levels in solution. 

Fourier transform infrared (FT-IR) spectra were obtained using a model Bio-RAD FTS-40 spectrophotometer in reflectance mode. Powdered samples were completely mixed with spectroscopic grade KBr at a 1:10 ratio. Multiple scans (*n* = 256) with resolutions of 4 cm^−1^ were obtained over 400–4000 cm^−1^ and corrected against potassium bromide as the background. ^1^H-nuclear magnetic resonance (NMR) spectra were recorded using a 3-channel Bruker Advance DRX 500 NMR spectrometer operating at 500.13 MHz using D_2_O/HCl (5% *v*/*v*) as the solvent. The following parameters were used: acquisition time = 10 μs, multiple scans (*n* = 64), and a 2 s recycle delay.

SEM images were recorded without gold coating with an electron microscope (Model SU8000, HI-0867-0003) with the following parameters: accelerating voltage (3000 V), increased magnification (5000×), and working distances (9.1 mm and 9.5 mm) for secondary electron and back scattered electron images, respectively. Energy-dispersive X-ray (EDX) spectra were obtained on a TEM model Hitachi HT7700 with Bruker Xflash 6t160 detector at fixed voltage (100 kV). Samples were prepared by depositing the powder onto a nickel grid. 

X-ray photoelectron spectroscopy (XPS) measurements were acquired using a Kratos (Manchester, UK) AXIS Supra system. The XPS system was equipped with a 500 mm Rowland circle monochromated Al K-α (1486.6 eV) source, combined hemi-spherical analyzer (HSA) and a spherical mirror analyzer (SMA). A hybrid slot with a fixed spot size (300 × 700 microns) was used. All survey scan spectra were collected in the −5 to 1200 binding energy range in 1 eV steps with a pass energy of 160 eV. High resolution scans of the two regions were also conducted using 0.05 eV steps with a pass energy of 20 eV. An accelerating voltage of 15 keV and an emission current of 15 mA were used for the data collection, where the binding energies were calibrated to C 1s (284.80 eV).

Ultraviolet visible (UV-Vis) spectral measurements were acquired with a Varian Cary 100 double beam spectrophotometer that used a 1 cm quartz cell at 23 °C between 250 and 550 nm. Fluorescence spectra were recorded on PTI (Photon Technology International) fluorimeter equipped with a xenon lamp source and 1 cm quartz cell at 23 °C. The spectra were recorded from 350 to 550 nm with the excitation wavelength at 330 nm. The slits width of the excitation and emission both were set to 3 mm. 

### 2.4. Sorption Studies

#### 2.4.1. Equilibrium Experiments

Adsorption experiments in batch mode were performed using 5 mg of adsorbent in a glass vial that was added to 6 mL of different initial concentration (*C_o_*) of Cu(II) solution from 50–800 mg L^−1^. Cu(II) solutions were made by dilution of a 1000 mg L^−1^ stock solution of CuSO_4_.5H_2_O. The vials were mixed in a horizontal shaker (SCILOGEX SK-O330-Pro) at 160 rpm for 24 h at 23 °C. After reaching equilibrium, the samples were filtered through 0.45 microfilter and diluted with HNO_3_ 2% *v*/*v*; the final concentration of Cu(II) in the filtrate (*C_e_*) was determined by FAA. Metal uptake of the adsorbent was calculated by Equation (1), where *C_o_* (mg L^−1^) and *C_e_* (mg L^−1^) are the concentrations of Cu(II) before and after the adsorption process.
(1)$$Q_{e}=\frac{\left(C_{o}-C_{e}\right)V\;}{m}$$

*Q_e_* (mg g^−1^) is the adsorbed Cu(II) at equilibrium condition, *V* (mL) is the volume of Cu(II) solution, and *m* (mg) is the adsorbent weight in each vial.

#### 2.4.2. Thermodynamic Studies

To evaluate the role of temperature on the adsorption process, the adsorbent was added to 100 mg L^−1^ Cu(II) solution in a glass vial and agitated at 160 rpm for 24 h at 288, 298, and 308 K. The standard enthalpy change (∆H°, kJ/mol), standard entropy change (∆S°, J/mol.K), and the standard Gibbs energy change (∆G°, kJ/mol) can be calculated using Equations (2)–(5).
∆G° = −RT ln K_s_(2)
(3)$$K_{s}\;=\;\frac{Q_{e}}{C_{e}}$$
∆G° = ∆H° − T∆S°(4)
(5)$$\ln\;K_{s}\;=\;(-\frac{\Delta H{}^{\circ}}{R})\;\frac{1}{T}\;+\;\frac{\Delta S{}^{\circ}}{T}$$

From the above equations, R is the gas constant (8.314 J/mol.K), *Q_e_* and *C_e_* are defined by Equation (1), and ∆H° and ∆S° values were calculated by the slope and intercept of a van’t Hoff plot (ln K_s_ vs. 1/T).

#### 2.4.3. Kinetic Experiments

Kinetic tests were carried out using a “one pot” method, with 20 mg L^−1^ (100 mL) as the initial concentration of Cu(II) solution at 23 °C. The adsorbent (100 mg) was wrapped in a fast filter paper (Whatman 2), which was saturated with Cu(II) solution before the start of the experiment. A 5 mL aliquot of the solution was removed at defined time intervals using a pipette. Samples were treated as equilibrium tests to determine the Cu(II) concentration. Kinetic studies were run by using 20, 50, and 100 mg L^−1^ of Cu(II) solution. The adsorption capacity versus time was evaluated using Equation (6).
(6)$$Q_{t}=\frac{\left(C_{o}-C_{t}\right)V\;}{m}$$

*Q_t_* (mg g^−1^) is the level of adsorbed Cu(II) per gram of adsorbent at variable time, *C**o* (mg L^−1^) is the initial concentration of Cu(II) solution, *C_t_* (mg L^−1^) is the concentration at variable time, *V* (L) is the volume of solution, and *m* (g) is the weight of the adsorbent.

#### 2.4.4. Isotherms and Modelling

The equilibrium isotherms where plotted by *Q_e_* vs. *C_e_* and analyzed using the Langmuir [31] (Equation (7)) and Sips [32] (Equation (8)) models. The Langmuir model accounts for homogenous (uniform site) and monolayer adsorption profiles, while the Sips model describes heterogeneous (multi-site) adsorption processes.
(7)$$Q_{e}=\frac{Q_{m}\;K_{L}\;C_{e}}{1+K_{L}\;C_{e}}$$
(8)$$Q_{e}=\frac{Q_{m}{\left(K_{S}\;C_{e}\right)}^{\;n_{s}}}{1+{\left(K_{s}C_{e}\right)}^{\;n_{s}}}$$

Here, *Q_e_* and *C_e_* are defined by Equation (1), *Q_m_* (mg g^−1^) is the maximum adsorption capacity, *K_L_* (L mol^−1^) is the Langmuir constant, *K_s_* (L mol^−1^) is the Sips constant, and n_s_ is the adsorbent surface heterogeneity parameter. A correlation coefficient (*R^2^*) was used to select the best-fit model.

The kinetic data were evaluated using the pseudo first order (PFO) [33], pseudo second order (PSO) [34], and intra-particle diffusion [35] models, as described by Equations (9)–(11).
(9)$$Q_{t}=Q_{e}\;\left(1-e^{k_{1}t}\right)$$
(10)$$Q_{t}=\frac{{Q_{e}}^{2}\;k_{2}t}{1+Q_{e}k_{2}t}$$
(11)$$Q_{t}\;=\;k_{i}t^{1/2}+C$$

*Q_t_* (mg g^−1^) is Cu(II) uptake at variable time and *Q_e_* (mg g^−1^) is adsorption capacity of the adsorbent at pseudo equilibrium conditions. The following rate constants: *k*_1_ (min^−1^), *k*_2_ (g mol^−1^ min^−1^), and *k_i_* (mg/g.min^1/2^), correspond to the respective adsorption models for the PFO, PSO, and intra-particle diffusion models, where *C* is a constant term. 

#### 2.4.5. Adsorbent Regeneration 

To test the reuse and regeneration of the adsorbent, ca. 5 mg of CS-Ac-An was added to Cu(II) solution (100 mg L^−1^; 6 mL) in a glass vial at 23 °C for 24 h. Subsequently, the remaining level of Cu(II) in the solution was determined by Equation (1). The metal loaded adsorbent was washed gently with water (3×) and regenerated by mixing with 5 mL of EDTA (0.01 M) for 3 h. After desorption and separation, the adsorbent was used for another 4 cycles to test its efficiency after each regeneration step.

### 2.5. UV-Vis and Fluorescence Spectroscopy

A stock solution of CS-Ac-An (1 g L^−1^) was prepared in acetic acid 2%. Copper solutions were made by the same procedure as described for the adsorption experiments. For UV-vis and quenching fluorescence experiments, solutions of CS-Ac-An (2 mL) were transferred to 3 mL quartz cuvettes and 0.5 mL of copper solutions with different concentrations were added by micropipette in a way that final Cu(II) concentrations were 0.005–5 mM. UV-Vis light absorption was measured between 250 and 550 nm. In the case of the measurement of fluorescence emission, the time interval between readings was 1 h. The Stern-Volmer equation (Equation (12)) was used to study the fluorescence quenching behavior of the fluorophore [36].
(12)$$\frac{F_{0}\;}{F}=1+K_{sv}\left[Q\right]$$

*F*_0_ and *F* are fluorescence emission intensity of the CS-Ac-An system at 420 nm in the absence and presence of Cu(II), where *K_sv_* (M^−1^) is the Stern-Volmer quenching constant and *Q* (M^−1^) is the concentration of Cu(II) as the quencher. 

## 3. Results and Discussion

### 3.1. Characterization

To confirm the grafting of aniline onto the backbone of chitosan (CS) through the acetaldehyde linker, an IR spectrum for pristine chitosan was recorded (Figure 1a), in agreement with IR reported previously (cf. Figure 2 in [30]). The strong OH and NH_2_ hydrogen band at 3000–3500 cm^−1^ disappeared upon grafting. Spectral bands near 1607 and 1496 cm^−1^ were attributed to C=C stretching of aromatic rings while C–H bending appeared at 749 cm^−1^. In Figure 1b, the ^1^H-NMR spectra for CS-Ac-An and chitosan were compared to confirm the grafting of chitosan. ^1^H-NMR signatures for the methyl and methine groups of the linker appear as a doublet (1.14–1.22 ppm) and a multiplet (2.10–2.22 ppm). Resonance lines of the arene protons appear between 7.01 and 7.26 ppm, in agreement with results from a previous report [30].

In Figure 2a, SEM results show that the smooth surface of CS undergoes a change upon grafting, as evidenced by porous and spongy surface features of CS-AC-An, in agreement with general trends noted for such surface grafting effects. After adsorption of Cu(II), the off white color of CS-Ac-An changed to blue and the adsorbed copper appeared as needles on the surface of adsorbent, as shown in Figure 2b [37]. The surface morphology reveals a notable change when comparing the SEM images before and after adsorption. The experimental results herein for CS and CS-Ac-An (without Cu(II)) concur with previously reported results [30]. EDX analysis was used to characterize the elemental composition of the flakes after Cu(II) adsorption (Appendix A and Appendix A). The results confirm the presence of copper and reveal that Cu(II) is not adsorbed solely as a precipitate of copper sulfate. The ratio of Cu(II) to S in copper sulfate is ~2:1, while EDX results showed the ratio of 4.2:1. In other words, copper is chelated through active sites of the adsorbent—mainly the sandwiching of Cu(II) between electron rich arene rings via cation-π interactions. 

### 3.2. Adsorption Mechanism

#### 3.2.1. FT-IR Spectroscopy

To study the bonding mode of Cu(II) and the nature of the interactions between the active sites and Cu(II) in solution, FT-IR spectra were obtained before and after adsorption (cf. Figure 1a). The intensity of broad hydrogen bond (3500 cm^−1^) decreased after metal adsorption which is an indication of Cu(II) coordination with –NH_2_ [38]. Notably, the bands at 1496 and 1607 cm^−1^ (C–C stretching vibrations in aromatic rings) disappeared completely, suggesting a strong interaction between Cu(II) and the pendant electron rich arene rings. Additionally, the band at 749 cm^−1^ (aromatic C–H bending) was broadened, appearing as a small shoulder after metal adsorption. Furthermore, a new band at 1545 cm^−1^ indicates that –NH_2_ was involved in metal adsorption. A new band appeared at 603 cm^−1^ in the spectrum after Cu(II) adsorption, which is attributed to the formation of N–Cu(II) bond [39]. Metal-ligand complex formation via Lewis acid-base interactions between the amine groups of chitosan with Cu(II), and a strong cation-π interaction (Cu(II)-arene), constitute the main adsorbent-adsorbate interactions. Based on FT-IR spectra, oxygen lone pairs do not contribute to the adsorption process.

#### 3.2.2. XPS Analysis

XPS analysis was used to verify the active sites between CS-Ac-An with Cu(II). Figure 3a illustrates the wide spectra of CS-Ac-An before and after copper adsorption. In Figure 3a, the binding energy (BE) of bands near 284 eV for C 1*s*, 396 eV for N 1*s,* and 532 eV for O 1*s* are clearly visible in both spectra. Furthermore, photoemission bands of Cu 2*p_3/2_* and Cu 2*p_1/2_* at 931.0 and 951.0 eV and two satellite peaks at 940.0 and 960.0 eV show that copper was adsorbed [40]. The high resolution XPS spectrum for Cu 2*p* is presented in Figure 3b. In order to assess the roles of potential functional groups in the adsorption process, the high resolution XPS results and deconvolutions of C *1s*, N *1s,* and O *1s* spectra before and after adsorption were analyzed (cf. Figure 3c–e). The binding energy and assignments are included in Appendix A. The high-resolution C*1s* spectrum can be deconvoluted into three bands before adsorption. The first, at 283.24 eV, is attributed to *sp^2^* hybridized carbon atoms in aromatic rings (–C=C–); the second, at 284.80 eV, is related to *sp^3^* hybridized carbon atoms (–C–C–); and the third, at 286.29, is attributed to carbon bonded to a heteroatom (i.e., –C–N–, –C–O–) [41,42]. After Cu(II) adsorption, the three bands did not change notably, but a newly formed peak at 287.29 eV is accounted for by cation-π interaction between Cu(II) and pendant aromatic rings of the adsorbent [43]. This interaction reduces the electron density of the ring by sharing electrons with Cu(II), resulting in a slightly higher binding energy. 

The XPS data of N 1*s* on CS-Ac-An before adsorption shows bands at 396.82 and 397.41 eV which relate to a free amine (–NH_2_) and a protonated amine group (–NH_3_^+^) with higher BE. After Cu(II) adsorption, a new peak at 399.05 eV was observed, suggesting the formation of N–Cu(II) bond, in which nitrogen lone pairs are shared with Cu(II) [44,45]. 

High resolution O1*s* spectra and its deconvoluted spectra show two bands at 529.81 and 530.48 eV before adsorption that are assigned to O–C and O–H signatures [42]. There is no obvious shift after copper adsorption that concurs with the notion that the lone pairs of O-heteroatoms are not primarily involved in the adsorption process. 

#### 3.2.3. ^1^H-NMR Spectroscopy

To provide further evidence that cation-π interactions occur in the adsorbent-adsorbate system, ^1^H-NMR spectra were recorded before and after adsorption process (cf. Figure 4). Three key changes occur in 6–9 ppm region after adsorption. ^1^H arene signatures are observed (7.11–7.46 ppm) that reveal a downfield shift to 7.14–7.50 ppm [46]. Complex formation with paramagnetic copper ion also broadens the well-resolved aromatic protons, suggestive of strong cation-π interactions [47], in agreement with the EDX results. Additionally, ^1^H nuclei of the amine group were not detectable for CS-Ac-An. However, the amine group was recognized after adsorption, further suggesting N–Cu(II) ligation contributes to lowering the proton exchange rate [48].

According to the FT-IR, XPS, and ^1^H-NMR observations, two main active sites are considered for CS-Ac-An, which include the amine groups of chitosan and the grafted aromatic ring (Figure 5). Cu(II) cations are coordinated between stacked aromatic rings and locked in their positions with the dative bonding of adjacent amine groups. A focused view is also shown in Appendix A. Sulfate anion adsorption likely occurs to maintain charge balance within the complex [49].

### 3.3. Adsorption Behavior

#### 3.3.1. Equilibrium

The uptake properties of CS-Ac-An were evaluated at variable Cu(II) levels in aqueous solution at ambient pH (6.5–7) and 23 °C. Figure 6 displays the experimental data and the best-fit results with the Langmuir and Sips isotherms (cf. Equations (7) and (8)), where the Sips model isotherm parameters for the adsorbents are listed in Table 1. By comparison, the Langmuir best-fit parameters are listed in Appendix A. According to the results in Figure 6, the adsorption capacity was raised by increasing the Cu(II) concentration to reach a plateau at high concentration. 

Adjusted correlation coefficient (Adj-*R^2^*) obtained for CS-Ac-An by the Sips model was higher than that for the Langmuir model. The Sips model adsorption capacity of CS-Ac-An is 106.6 mg g^−1^, increased by ca. 20-fold relative to pristine chitosan (cf. Figure 6). The value of *Q_m_* obtained is notably higher in comparison with other reported studies (Appendix A) [20,21,22,50,51,52,53,54,55,56]. According to the results, CS-Ac-An is described as an adsorbent with incremental surface heterogeneity which is predictable from its chemical structure relative to chitosan [57]. The relatively high adsorption capacity can be accounted for by the synergism that results from the combined effects of strong (cation-lone pair) and weak (cation-π) interactions.

#### 3.3.2. Temperature Effects

The role of temperature in Cu(II) removal from water at equilibrium conditions is shown in Appendix A. It can be seen that the adsorption capacity (*Q_e_*) increased from 27.8 to 77.5 mg/g for temperatures from 15 to 35 °C. This is in accordance with other case studies for metal ion uptake by modified chitosan adsorbents [58,59,60,61]. Thermodynamic parameters are calculated using Equations (2)–(5) and are shown in Appendix A and Appendix A. Positive values of ∆H° characterize an endothermic adsorption process while a negative value of ∆G° illustrates the favorable (spontaneous) energetics for the uptake process in the on CS-Ac-An/Cu(II) system. The positive value for ∆S° likely indicates desolvation of the Cu(II) species upon adsorption by CS-Ac-An.

#### 3.3.3. Kinetic Results

Adsorption kinetic profiles are well-described by fitting the results to a pseudo-second order (PSO) model (Equation (10)). The fitting parameters for the PFO and PSO models are shown in Figure 7a and Appendix A. The Adj-*R*^2^ correlation coefficient is somewhat higher for the PSO kinetic model relative to those determined for PFO model (Equation (9)). Thus, it is concluded that the PSO model provides a more reliable best-fit to the experimental data relative to the PFO model. The adsorption capacity is shown to undergo a rapid increase within the first 200 min that reaches ca. 80% of its maximum value due to the large number of available active adsorption active sites in this region. By increasing the contact time, the active sites of the adsorbent undergo saturation, whereas the adsorption capacity does not change appreciably. 

Since the PSO or PFO models do not provide an account of the adsorption rate determining step (RDS), a third model was employed, known as the intra-particle diffusion or Weber and Morris model (Equation (11)) [62]. According to this model, the adsorption of Cu(II) is described by two key steps: i) transport of cations to the boundary layer toward the surface of the adsorbent (external surface adsorption) and ii) transport of ions from the surface into the adsorbent pores, referred to as intraparticle diffusion [63]. A plot of *Q_t_* vs. *t*^1/2^ for three different initial concentrations of Cu(II) is shown in Figure 7b. The initial step would account for the external surface adsorption, while the final step demonstrates the role of intraparticle diffusion. The values of *k_i_* that relate to each respective region are shown in Appendix A. The *k*_1_ values for the first step decrease with an increase of the initial Cu(II) concentration, which indicates that the external surface adsorption is the RDS at higher Cu(II) concentration. The role of inter-ionic interactions between positively charged Cu(II) ions accounts for a lower potential to penetrate into the boundary layer while lowering the initial Cu(II) concentration. This effect can result in more rapid penetration into the boundary layer without repulsions that contribute to lowering of the RDS for the second step or intra-particle diffusion contributions. According to this graph, the rate limiting step (RDS) of adsorption process is largely dependent on the level of Cu(II) species.

#### 3.3.4. Adsorbent Regeneration 

Regeneration of spent adsorbents is a key consideration from the viewpoint of practical field applications. To evaluate the reusability of Cu(II)-loaded CS-Ac-An, desorption experiments were carried out using an EDTA solution [64]. After each desorption step, the removal of Cu(II) is shown by the loss of blue color from the bound state of the adsorbent, as compared with the off-white color appearance before adsorption (*cf.* inset of Figure 2). The adsorption-desorption process was continued for five cycles; the results are shown in Appendix A. The adsorption capacity (*Q_e_*) decreased by 0.5% in the second cycle, 5.5% in the third cycle, 7.9% in the fourth cycle, and 9.7% in the fifth cycle. While *Q_e_* decreases slightly over the adsorption-desorption cycles, the CS-Ac-An adsorbent represents a promising material for Cu(II) removal for wastewater purification due to its relatively high adsorption capacity. 

### 3.4. Fluorescence Studies

Metal-ion sensing behavior of CS-Ac-An toward Cu(II) was studied using the fluorescence emission properties of the modified chitosan. While the adsorption spectra do not show any substantial change in the presence of Cu(II) species (Appendix A), the fluorescence emission of CS-Ac-An undergoes notable quenching [65,66,67,68,69]. Different excitation wavelengths were tested and showed a trend for the λ_ex_-dependent-λ_em_ behavior. In other words, the emission maxima revealed a red-shift as the wavelength of the excitation increased. The excitation at 330 nm shows the strongest photoluminescence emission band at 420 nm (Appendix A) [68]. The pristine CS-Ac-An sample has an intense fluorescence emission band that red shifts upon Cu(II) coordination, from 420 to 430 nm (Figure 8a). Analysis of the fluorescence quenching data by the Stern-Volmer equation in the absence and presence of Cu(II) yields a quenching constant of 56 M^−1^ (Figure 8b) [36,69]. This graph represents that the emission diminishes rapidly with increasing levels of Cu(II), whereas this strong effect becomes attenuated at more elevated Cu(II) levels. It is possible to determine the Cu(II) concentration using this graph in the range of 0.05 to 5 mM (3–300 mg L^−1^) with a correlation coefficient (*R*^2^) of 0.997 [70]. 

In this case, fluorescence quenching is accounted for by a shift of the π-electron density toward the Cu(II) cation which reduces the electron density in the HOMO of the arene moieties. In the other words, Cu(II) as the guest species bind to the fluorophore and quenches its fluorescence emission to yield a chelation enhanced fluorescence quenching (CHEQ) sensor [36]. 

As noted in Figure 8b, a reduction in the fluorescence emission (by 14%) was observed when the concentration of Cu(II) in solution reached 0.3 mg L^−1^. This concentration is considerably lower than the maximum allowable level of copper in drinking water [8]. Thus, CS-Ac-An *molecular-tweezers* can be deployed as a responsive adsorbent with utility for in situ detection of Cu(II) with high sensitivity. 

## 4. Conclusions

A grafted form of chitosan (CS-Ac-An) was fabricated and structurally characterized by FT-IR and ^1^H-NMR spectroscopy. The adsorption properties of CS-Ac-An in the presence of Cu(II) were studied in aqueous solution at ambient pH (6.5–7) and temperature. The Sips isotherm model accounts for the equilibrium adsorption properties, where a relatively high adsorption capacity (106.6 mg g^−1^) was noted (>20-fold) relative to chitosan. The kinetic uptake profiles are well-described by the PSO model; the intra-particle diffusion model was applied to reveal the rate limiting step of the process. Temperature dependent adsorption studies reveal that the process is entropy-driven, according to changes in hydration of Cu(II) upon adsorption. The adsorption process reveals that the pendant aromatic rings and amine groups serve as the active adsorption sites that contribute synergistically due to the combined effect of weak (cation-π) and strong (dative) interactions, respectively. These metal-ligand bonding processes are the prominent metal-adsorbent interactions, as supported by the spectral results herein. SEM images clearly reveal porous surface features of the adsorbent before adsorption and copper aggregates after uptake, where the surface morphology changed during the adsorption process. EDX analyses reveal that Cu(II) ions are adsorbed mainly by active sites of the adsorbent and not as copper sulfate. Five cycles of adsorption-desorption show CS-Ac-An can be used repeatedly as an adsorbent without a significant loss of its adsorption capacity (<10%), in line with the chemisorption nature of the process. In aqueous media, the adsorbent exhibits fluorescence emission that undergoes quenching in the presence of low levels of Cu(II), where a linear response over a nominal range allows for practical in situ detection of Cu(II) that is relevant to drinking water guidelines. The dual properties of CS-Ac-An allow for its utility as a responsive material for the detection of Cu(II), along with its unique solid phase extraction properties and relatively high adsorption capacity.

## Figures and Tables

**Figure 1 molecules-25-01052-f001:**
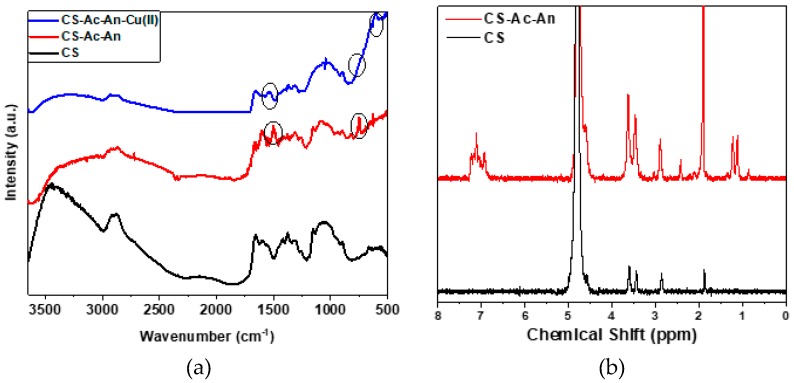
(**a**) FT-IR spectra of pristine chitosan (CS), CS-Ac-An, and CS-Ac-An-Cu(II) (after adsorption). (**b**) ^1^H-NMR spectra of pristine chitosan (CS) and CS-Ac-An (before adsorption).

**Figure 2 molecules-25-01052-f002:**
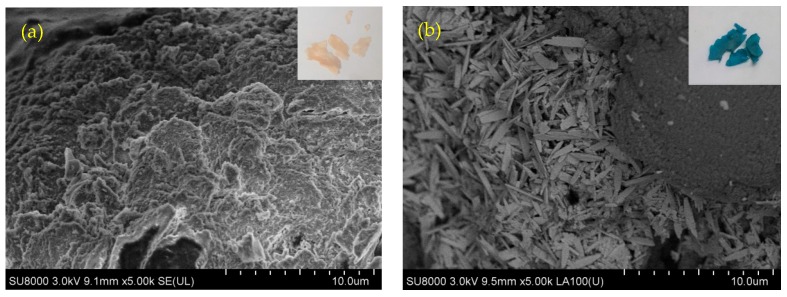
SEM images: (**a**) Secondary electron image of CS-Ac-An (before adsorption). (**b**) Backscattered electron image of CS-Ac-An-Cu(II) with needle-shaped copper sulfate on the surface of the adsorbent (concentration of Cu(II) solution: 100 mg L^−1^). Insets: images that illustrate the off-white color of adsorbent ((a) before adsorption) and the deep blue color of adsorbent ((b) after Cu(II) adsorption).

**Figure 3 molecules-25-01052-f003:**
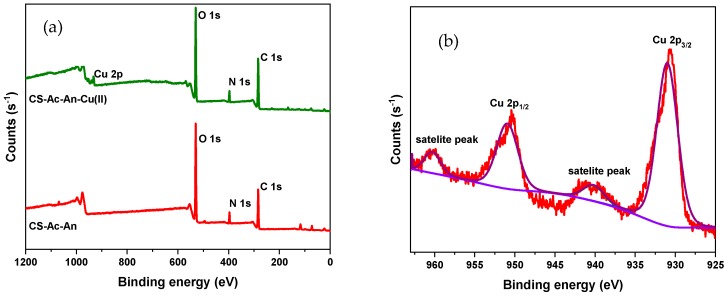
XPS spectra of (**a**) CS-Ac-An before and after metal adsorption; (**b**) Cu 2*p* high resolution spectra; (**c**) C 1*s* high resolution spectra before and after adsorption; (**d**) N 1*s* high resolution spectra before and after adsorption; and (**e**) O 1*s* high resolution spectra before and after adsorption.

**Figure 4 molecules-25-01052-f004:**
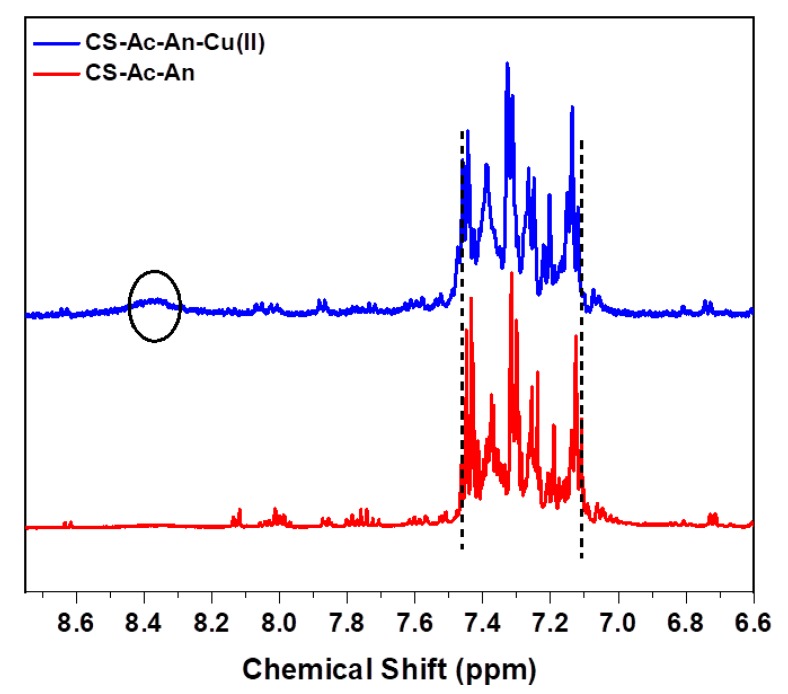
^1^H-NMR spectra before and after adsorption. Downfield shifts and spectral broadening of the aromatic region upon formation of a N–Cu(II) complex is indicated.

**Figure 5 molecules-25-01052-f005:**
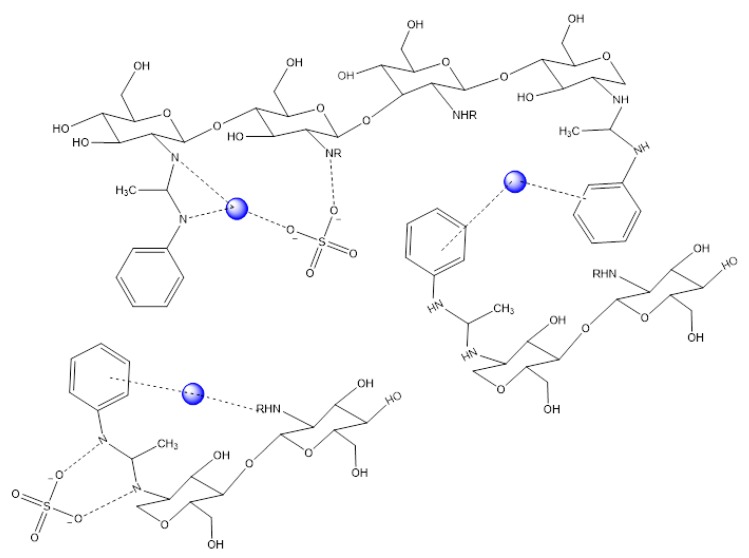
CS-Ac-An structure with interactions (dotted lines) between Cu(II) (blue circles) ions and active sites of the adsorbent, where the solvent is omitted for clarity purposes.

**Figure 6 molecules-25-01052-f006:**
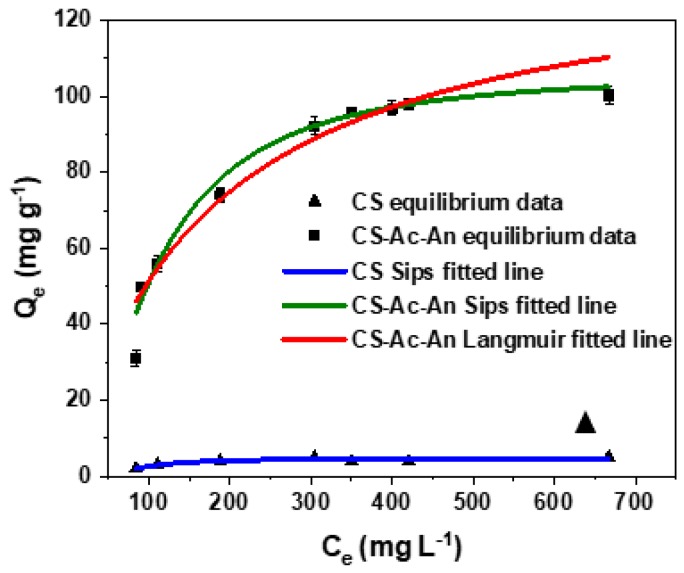
Adsorption isotherms for chitosan (▲) fitted to the Sips model and CS-Ac-An (■) fitted to the Langmuir and Sips models at ambient pH (6.5–7) and 23 °C. Cu(II) concentration: 50–800 mg L^−1^, contact time: 24 h, adsorbent dosage: 5 mg. The adsorption profile for chitosan is shown for comparison.

**Figure 7 molecules-25-01052-f007:**
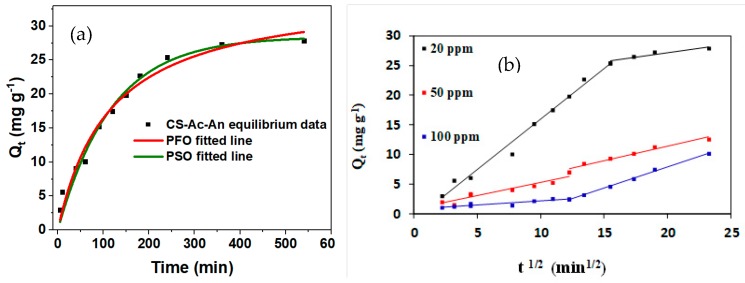
(**a**) Kinetic adsorption profile of Cu(II) uptake with CS-Ac-An at ambient pH (6.5–7) and 23 °C that were fit by the PFO and PSO models. Cu(II) concentration: 20 mg L^−1^, volume: 100 mL, and adsorbent dosage: 100 mg. (**b**) Kinetic data analyzed by the Weber and Morris model for variable Cu(II) concentration at ambient pH (6.5–7) and 23 °C. Cu(II) concentration: 20, 50, and 100 mg L^−1^; volume: 100 mL; and adsorbent dosage: 100 mg.

**Figure 8 molecules-25-01052-f008:**
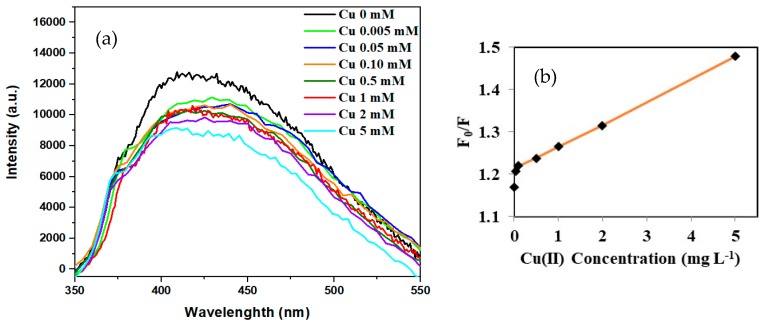
(**a**) Fluorescence spectra of CS-Ac-An in the presence of variable Cu(II) levels in aqueous acetic acid (2 wt%), λ_ex_ = 330 nm. (**b**) Plot of F_0_/F against Cu(II) concentration (Stern-Volmer equation).

**Table 1 molecules-25-01052-t001:** Sips model adsorption parameters for Cu(II) uptake with CS and CS-Ac-An at 23 °C.

CS				CS-Ac-An			
Q_m_ (mg g^−1^)	K_s_ (L g^−1^)	n_s_	Adj-R^2^	Q_m_ (mg g^−1^)	K_s_ (L g^−1^)	n_s_	Adj-R^2^
4.56	11.2	2.88	0.851	106.6	9.52	1.85	0.978

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
