# Peer review of "Cu(II) Ion Adsorption by Aniline Grafted Chitosan and Its Responsive Fluorescence Properties"

_molecules, 2020, doi:10.3390/molecules25051052_

Round 1

Reviewer 1 Report

As removal of heavy metal species in aquatic environments is actual problem today the contents presented in paper “Cu (II) Ion Adsorption by Aniline Grafted Chitosan and its Responsive Fluorescence Properties” should be attractive to the readers of the Molecules journal. This pollution process is very actual problem today worldwide in different areas.

The paper is easy to read, the experimental procedure is satisfactory, and a grafted form of chitosan (CS-Ac-An) was successfully fabricated and well characterized, as well as applied. Reference list is up to date and more than adequate

I can recommend publication of the following manuscript with minor revision.

 The following are some comments: 

Results and discussions

The recommendation is to make it easier to track the results of the FTIR spectra: or to separate them into two figures (one in the characterization and the other in the Adsorption Mechanism section) or to merge the discussion. Adsorption behavior and Kinetics:

The authors used adsorbent dosage of 5 mg.

It is known that with such small quantities a large error is obtained, i.e. a real state of the system is not obtained. The kinetics were done at adsorbent dosage of 100 mg.

It takes that the pollutant concentration, the volume and mass of the adsorbent (i.e. liquid to solid phase ratio) to be approximately the same to obtain a realistic state of the system. Can the authors repeat the kinetics at the same adsorbent dosage?

Please define ambient pH

Author Response

Author Response to Reviewer Report on MS ID:  molecules-723206

 Reviewer Report #1

  1. The recommendation is to make it easier to track the results of the FTIR spectra: or to separate them into two figures (one in the characterization and the other in the Adsorption Mechanism section) or to merge the discussion:

Response: It should be noted that the preparation of the CS-Ac-An was reported previously in ref. [30]; see “Vafakish, B.; Wilson, L.D. Surface Modified Chitosan: An Adsorption Study of a Biopolymer “Tweezer-Like” System with Fluorescein. Surfaces 2019, 1–27.”

Therefore, we did not report a detailed discussion of the characterization results. In light of the previous report, we did not see the need to add one more figure to the manuscript since the the main focus of the present work relates to the  adsorption mechanism. The readers are referred to the previously reported results in ref. [30]. Instead, the FT-IR spectra before and after adsorption by the aniline grafted chitosan (CS-Ac-An), as discussed herein (see Section 3.1).

  1. The authors used adsorbent dosage of 5 mg. It is known that with such small quantities a large error is obtained, i.e. a real state of the system is not obtained. 

Response: We agree with the reviewer that 5 mg in 5 mL is not directly comparable with typical conditions used for practical adsorption systems. However, the relatively low dosage and high uptake reflects the “contributions” of the present work, where the adsorbent displays relatively high copper adsorption in spite of the relatively low adsorbent dosage.

Independently, we tested the adsorption thermodynamics at conditions with up to 15 mg/20 mL (3-4 folder lower adsorbent dosage), where it was observed that the adsorption capacity was unaffected­­. The error alluded to above was minimized by the use of a high precision balance (±0.01 mg), as evidenced by the negligible error bars on the graph. Notwithstanding, we would like to report on the high uptake capacity even at the adsorbent dosages herein.

  1. The kinetics were done at adsorbent dosage of 100 mg. It takes that the pollutant concentration, the volume and mass of the adsorbent (i.e. liquid to solid phase ratio) to be approximately the same to obtain a realistic state of the system. Can the authors repeat the kinetics at the same adsorbent dosage?

Response: With low amount of the adsorbent, the chance of adsorbent-adsorbate interaction is less so reaching the plateau takes longer time. The kinetic experiments were obtained at variable dosage, where 50 and 100 mg of adsorbent was used. The latter dosage (1 mg/mL; 1:1 solid-liquid ratio) reveals the best results and are reported in the manuscript, in line with the greater availability of adsorption sites.

  1. Please define ambient pH

Response: The corresponding revision was carried out.

The authors wish to acknowledge reviewer #1 for the insightful and constructive comments. We feel that such recommendations have improved the overall quality of this submission and we have carried out further editing of language and syntax throughout to further enhance the clarity and to meet the high standards of this journal.

Reviewer 2 Report

Dear Authors,

The manuscript entitled "Cu(II) Ion Adsorption by Aniline Grafted Chitosan and its Responsive Fluorescence Properties" is very interesting, it describes in detail the research results, its analysis and conclusions. In particular, very careful analysis of the experimental results, very precise mathematical fitting and careful conclusions make the manuscript of high scientific value.

I recommend this paper to publish in present form.

Author Response

Author Response to Reviewer Report on MS ID:  molecules-723206

 Reviewer #2

Dear Authors,

The manuscript entitled "Cu(II) Ion Adsorption by Aniline Grafted Chitosan and its Responsive Fluorescence Properties" is very interesting, it describes in detail the research results, its analysis and conclusions. In particular, very careful analysis of the experimental results, very precise mathematical fitting and careful conclusions make the manuscript of high scientific value.

I recommend this paper to publish in present form.

 Response:  The authors wish to acknowledge reviewer #2 for appraisal of this manuscript. We have carried out further editing of language and syntax throughout to further improve the clarity and to meet the high standards of this journal.